# Characteristics of Non-Smokers' Exposure Using Indirect Smoking Indicators and Time Activity Patterns

**Byung Lyul Woo [1,2], Min Kyung Lim [3], Eun Young Park [4], Jinhyeon Park [5] , Hyeonsu Ryu [1], Dayoung Jung [1], Marcus J. Ramirez [2] and Wonho Yang [1,5,\*]**

[1] Department of Occupational Health, Daegu Catholic University, Gyeongsan-si, Gyeongbuk 38430, Korea; yissoyi@gmail.com (B.L.W.); clover92@cu.ac.kr (H.R.); ekdud37@naver.com (D.J.)

[2] Industrial Hygiene, Preventive Medicine, Force Health Protection, U. S. Army Medical Department Activity-Korea/65th Medical Brigade, Unit # 15281, APO AP 96271-5281, USA; marcusram2003@gmail.com

[3] Department of Cancer Control and Population Health, National Cancer Center, Graduate School of Cancer Science and Policy, Goyang-si, Gyeonggi-do 10408, Korea; mickey@ncc.re.kr

[4] Division of Cancer Prevention & Early Detection, National Cancer Control Institute, National Cancer Center, Goyang-si, Gyeonggi-do 10408, Korea; goajoa@ncc.re.kr

[5] Center of Environmental Health Monitoring, Daegu Catholic University, Gyeongsan-si, Gyeongbuk 38430, Korea; venza11@naver.com

\* Correspondence: whyang@cu.ac.kr

**Abstract:** Since the global enforcement of smoke-free policies, indoor smoking has decreased significantly, and the characteristics of non-smokers' exposure to secondhand smoke (SHS) has changed. The purpose of this study was to assess the temporal and spatial characteristics of SHS exposure in non-smokers by combining questionnaires and biomarkers with time activity patterns. To assess SHS exposure, biomarkers such as cotinine and 4-(methylnitrosamino)-1-3-(pyridyl)-1-butanol (NNAL) in urine and nicotine in hair were collected from 100 non-smokers in Seoul. Questionnaires about SHS exposure and time activity patterns were also obtained from the participants. The analysis of biomarker samples indicated that about 10% of participants were exposed to SHS when compared with the criteria from previous studies. However, 97% of the participants reported that they were exposed to SHS at least once weekly. The participants were most exposed to SHS in the outdoor microenvironment, where they spent approximately 1.2 h daily. There was a significant correlation between the participants' time spent outdoors and self-reported SHS exposure time ($r^2 = 0.935$). In this study, a methodology using time activity patterns to assess temporal and spatial characteristics of SHS exposure was suggested. The results of this study may help develop policies for managing SHS exposure, considering the time activity patterns.

**Keywords:** secondhand smoke; time activity; smoke-free; exposure assessment

## 1. Introduction

Secondhand smoke (SHS) is defined as a mixture of the smoke from the burning of tobacco products and smoke exhaled by smokers [1]. SHS exposure can cause adverse health effects such as respiratory disease, cardiovascular disease, and lung cancer [2,3]. In particular, exposure to SHS has become an important health issue due to its association with sudden infant death syndrome (SIDS) [4,5]. The World Health Organization (WHO) has estimated that, globally, more than 6 million people die from smoking, and the mortality rate from SHS exposure has reached 900,000 [6,7].

The WHO has organized the Framework Convention on Tobacco Control (FCTC) and encourages countries to implement education and awareness programs regarding the risks of SHS with the aim of a "Smoke Free World" [8]. According to the FCTC, fifteen developed and developing countries have successfully implemented smoke-free policies in indoor environments such as homes, workplaces, and public places during the last decade [9,10].

In Europe, under the theme of "lifting the smokescreen", the importance of SHS monitoring was recognized, and the results are being utilized to assess smoke-free policies [11]. In Canada, the Smoking Regulatory Index (a new way to measure public health performance) was developed and utilized to assess as an indicator of how well people are protected from SHS and the effectiveness of the policies to minimize SHS exposure [12]. Since the enactment of the National Health Promotion Act in Korea in 1995, indoor non-smoking areas have been gradually expanded, while a few outdoor areas have been designated as non-smoking areas through revisions of the corresponding laws in 2010 and 2012 [13,14]. As a result, the exposure rate of non-smokers to indoor SHS has significantly decreased [15].

However, as most indoor environments were designated as non-smoking areas, smokers had to shift their designated smoking areas from indoors to outdoors [16]. Consequently, non-smokers are more likely to be exposed to SHS at entrances or terraces of buildings with a smoking ban [17]. Hence, several countries are expanding non-smoking areas to outdoor locations of hospitals, kindergartens, restaurants, sports facilities, and public transportation [18,19]. However, smoking is still prevalent at the entrances of buildings such as restaurants, bars, shops, and universities in Korea [13]. Sureda et al. investigated the concentration of particulate matter with diameter less than 2.5 μm ($PM_{2.5}$) as an indicator of SHS at entrances and corridors of buildings with a smoking ban and compared the results, which showed that there was no significant difference in concentrations at either of the two places [20].

Previous studies have mainly used questionnaires and biological sampling for SHS indicators [21,22]. However, these methods provide limited information on specific SHS exposure duration and microenvironments. Therefore, this study attempted to understand the temporal and spatial characteristics of SHS exposure using time activity patterns, and evaluated the relationship between these characteristics and the biological sampling results.

## 2. Materials and Methods

### 2.1. Selection of Study Participants

Among the participants recruited for the basic SHS exposure survey conducted by the National Cancer Center, 100 non-smokers aged 19 to 74 and residing in Seoul were randomly selected. Questionnaires, time activity diaries, and biomarker samples were collected from April to May 2017. Past smokers who had quit smoking for more than 6 months and those who had not dyed their hair within 3 months were included. On the first day, the survey consent form and questionnaire were filled out along with collection of the first urine and hair samples. The participants were required to record their time activity diary for 7 days. On the seventh day, the time activity diary and the second urine samples were collected. This study was approved by the National Cancer Center for IRB vide approval number NCC2017-0074.

### 2.2. Questionnaire and Time Activity Diary

The questionnaire included demographic characteristics such as the subject's gender, age, occupation, housing type, and income, exposure characteristics to SHS such as places where the participants were mainly exposed to SHS, time spent in different microenvironments, the number of smokers within their family, and their viewpoint on the smoking policy. The used questionnaire was adapted from the previous studies [15,23–27]. In addition, the records of the participants' time activity patterns were collected to assess the exposure. The participants' activities and microenvironments

were recorded every 30 min [28]. The number of smokers observed, the number of cigar butts found, and the number of self-reported exposure to SHS in the microenvironments were also recorded.

### 2.3. Cotinine

Cotinine has a half-life of 18–24 h, which means that the cumulative exposure to SHS can last 2–3 days [29–31]. In addition, cotinine is often used as an index of SHS exposure as internal cotinine concentration is rarely affected by other factors [32]. High-performance liquid chromatography tandem mass spectrometry (HPLC-MS/MS, Agilent 1100 series, and API 4000, AB Sciex, Santa Clara, USA) was used for analysis [33].

### 2.4. NNAL

NNAL (4-(methylnitrosamino)-1-3-(pyridyl)-1-butanol, a metabolite of 3-pyridyl-1-butanone) can be found in the urine of non-smokers exposed to SHS, and the half-life of NNAL is approximately 1 to 2 weeks. [21]. As NNAL and NNAL-Gluc are biomarkers of lung carcinogens released from cigarettes, they are not only extremely useful indicators, but have also been reported to be superior to other biomarkers in the comprehensive assessment of various biomarkers related to SHS [34]. Liquid chromatography tandem mass spectrometry (LC-MS/MS; Agilent 1260 series & Triple Quadrupole 5500: Turbo Ion Spray TM source, AB Sciex, Santa Clara, USA) was used for the NNAL analysis [33].

### 2.5. Nicotine in Hair

Hair is a stable specimen that is easy to collect and is hardly metabolized by drugs. As its growth rate is slow, hair reflects relatively long-term exposure to chemicals, and approximately 1 cm of hair reflects SHS exposure of approximately a month [35]. LC-MS/MS was used to analyze nicotine in hair samples [36].

### 2.6. Criteria for Biomarkers to Assess SHS Exposure

The criteria for classifying exposure to SHS of non-smokers were established by reviewing literature and articles suggested by several organizations [21,37,38]. The non-smokers were classified to be exposed SHS when the concentration above the limit of quantitation (LOQ) was detected as the results of biomarker analysis, or according to cut-off concentration (1 ng/mL, 3.77 pg/mL, and 2 ng/mg, cotinine in urine, NNAL in urine and nicotine in hair respectively). The limits of quantitation (LOQ) of cotinine, NNAL, and nicotine were obtained according to the Clinical & Laboratory Standards Institute (CLSI) guidelines.

### 2.7. Statistical Analysis

Correlation was used to compare the time activity patterns, concentrations of biomarkers, number of cigarette butts found, number of smokers observed, and the number of exposures to SHS. The results of correlation analysis were presented by coefficient of correlation ($R^2$), and statistical analysis was performed using IBM SPSS (Version 19).

## 3. Results

### 3.1. Characteristics of Participants and SHS Exposure at Microenvironments

Table 1 shows the demographic characteristics of the participants. All subjects were distributed uniformly by gender, age, and occupation. Of these, 16% were unemployed, 87% had an education level of college graduates or higher, 69% were married, and 70% resided in apartments or multi-family houses. Additionally, 27% were past smokers who had quit smoking, and 22% of the participants drank more than twice a week. A total of 97% participants reported that they were exposed to SHS at least once a

week and that they experienced SHS exposure outdoors most of the time (92%). The SHS exposure at public places was 62%, which was higher than at home (36%) and workplaces (39%).

**Table 1.** General characteristics and major microenvironments where secondhand smoke (SHS) exposure occurred.

| Characteristics | N | Home (%) | Workplace (%) | Public Place (%) | Boundary [1] (%) | Outdoor (%) |
|---|---|---|---|---|---|---|
| Gender | | | | | | |
|   Male | 50 | 15 (30) | 23 (46) | 32 (64) | 40 (80) | 45 (90) |
|   Female | 50 | 21 (42) | 16 (32) | 30 (60) | 39 (78) | 47 (94) |
| Age | | | | | | |
|   20–29 | 17 | 6 (35) | 6 (35) | 13 (76) | 15 (88) | 16 (94) |
|   30–39 | 25 | 8 (32) | 14 (56) | 18 (72) | 21 (84) | 24 (96) |
|   40–49 | 23 | 7 (30) | 10 (43) | 12 (52) | 18 (78) | 21 (91) |
|   50–59 | 16 | 7 (44) | 4 (25) | 11 (69) | 12 (75) | 15 (94) |
|   60 ≤ | 19 | 8 (42) | 5 (26) | 8 (42) | 13 (68) | 16 (84) |
| Job | | | | | | |
|   Supervisory/profession | 24 | 11 (46) | 11 (46) | 15 (63) | 19 (78) | 24 (100) |
|   Office worker | 23 | 8 (35) | 9 (39) | 15 (65) | 19 (83) | 20 (87) |
|   Service/retail | 21 | 6 (29) | 13 (62) | 14 (67) | 19 (90) | 21 (100) |
|   Technician/labor | 16 | 5 (31) | 6 (38) | 8 (50) | 11 (69) | 13 (81) |
|   Unemployed | 16 | 6 (38) | 0 (0) | 10 (63) | 11(69) | 14 (88) |
| Education | | | | | | |
|   High school or lower | 13 | 5 (38) | 5 (38) | 6 (46) | 9 (69) | 11 (85) |
|   College or higher | 87 | 31 (36) | 34 (39) | 56 (64) | 70 (80) | 81 (93) |
| Household income [2] | | | | | | |
|   <USD 30,000 | 7 | 3 (43) | 3 (43) | 5 (71) | 5 (71) | 7 (100) |
|   USD 30,000–60,000 | 31 | 10 (32) | 11 (35) | 20 (65) | 22 (71) | 27 (87) |
|   USD 60,000–80,000 | 22 | 10 (45) | 8 (36) | 15 (68) | 19 (86) | 20 (91) |
|   >USD 80,000 | 23 | 7 (30 | 11 (48) | 14 (61) | 18 (78) | 22 (96) |
| Marital status | | | | | | |
|   Married | 69 | 25 (36) | 25 (36) | 40 (58) | 53 (77) | 64 (93) |
|   Not married | 31 | 11 (35) | 14 (45) | 22 (71) | 26 (84) | 28 (90) |
| Residential type | | | | | | |
|   Single-family home | 5 | 1 (20) | 1 (20) | 2 (40) | 3 (60) | 5 (100) |
|   Apartment | 70 | 25 (36) | 25 (36) | 45 (64) | 55 (79) | 62 (89) |
|   Townhouse | 20 | 8 (40) | 10 (50) | 12 (60) | 18 (90) | 20 (100) |
|   Multi-purpose building [3] | 5 | 2 (40) | 3 (60) | 3 (60) | 3 (60) | 5 (100) |
| Past smoker | | | | | | |
|   Yes | 27 | 12 (44) | 12 (44) | 19 (70) | 23 (85) | 25 (93) |
|   No | 73 | 24 (33) | 27 (37) | 43 (59) | 56 (77) | 67 (92) |
| Alcohol consumption | | | | | | |
|   Never | 22 | 7 (32) | 4 (18) | 13 (59) | 17 (77) | 19 (86) |
|   once/month | 28 | 12 (43) | 11(39) | 19 (68) | 21 (75) | 26 (93) |
|   once/week | 28 | 9 (32) | 12 (43) | 18 (64) | 22 (78) | 26 (93) |
|   ≥twice/week | 22 | 8 (36) | 12 (55) | 12 (55) | 19 (86) | 21 (95) |
| Total | 100 | 36 (36) | 39 (39) | 62 (62) | 79 (79) | 92 (92) |

[1] Boundary: entrance, porch, terrace, top of building, parking lot, and so on. [2] Thirteen unemployed and one respondent did not answer to this question. [3] Mixed use buildings, e.g., residential and commercial apartment, office, and hotel.

### 3.2. SHS Exposure Duration at Microenvironments

Figure 1 shows the duration of SHS exposure for each microenvironment. The participants reported that they were exposed to SHS for an average of 4.5 h during a week, and the exposure duration was longer indoors than it was outdoors. The SHS exposure duration was the longest in indoor workplaces and shortest in indoor public places. There was no statistically significant difference in exposure time by microenvironment. The SHS exposure duration was longer in restaurants, followed by bars and transportation-related facilities.

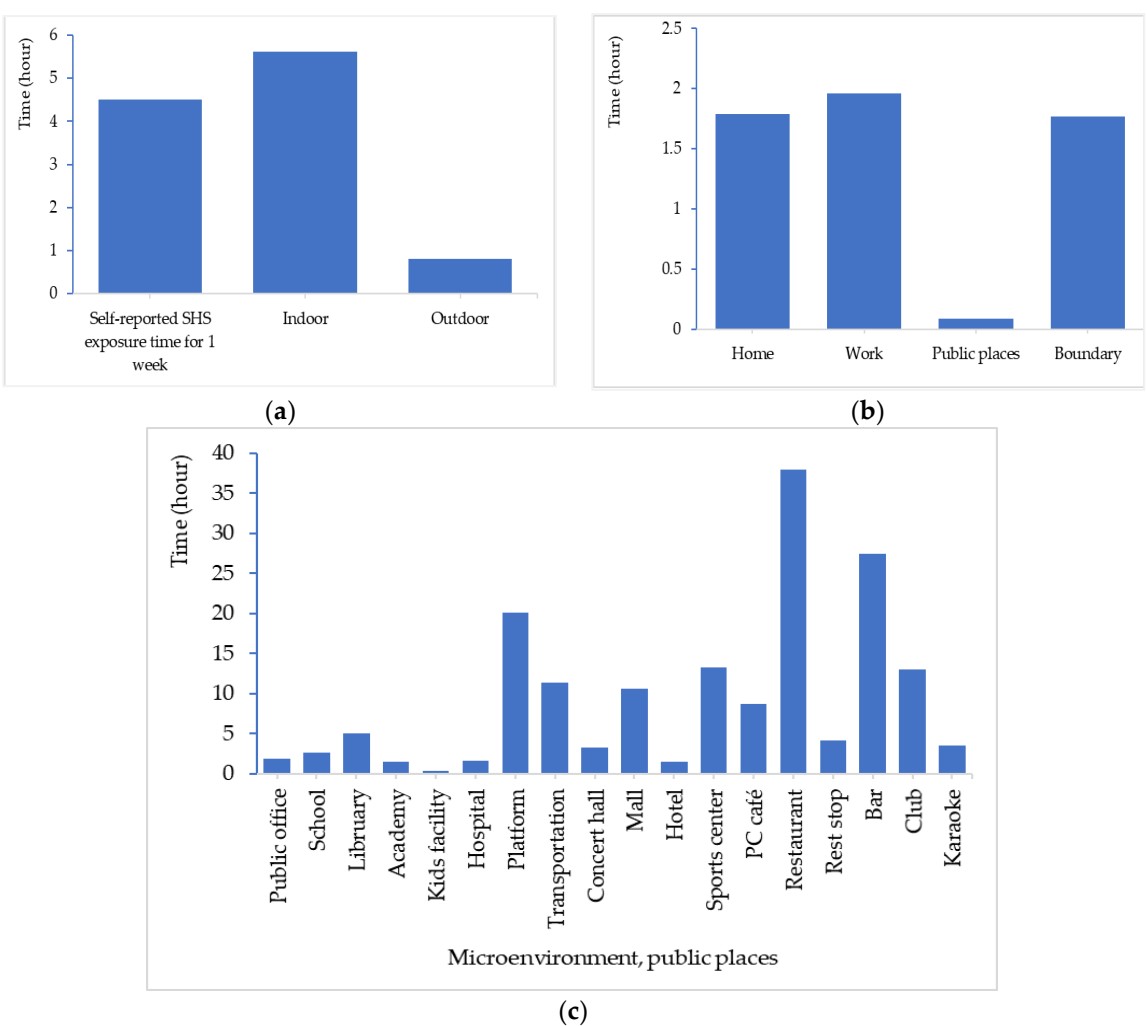

**Figure 1.** SHS exposure duration: (**a**) average for a week, indoor and outdoor; (**b**) home, workplace, public place, and boundary; and (**c**) public places.

### 3.3. Time Activity Pattern

The microenvironments where the participants stayed and the time spent in each microenvironment during a week are shown in Table 2. The participants spent most of their time indoors and spent approximately 1.2 h outdoors per day. The participants' time spent at the workplace and public transportation for weekdays was relatively higher than on weekends. In contrast, the participants spent more time at home, and the duration of private car use increased during the weekend.

Frequency distributions of the time spent in each microenvironment for weekdays and weekends are shown in Figure 2. The participants stayed at home most of time, and their working hours were approximately from 7 am to 6 pm during weekdays. The rate of visiting and using restaurants peaked at noon, and time spent outdoors increased at 8 a.m., 12 p.m., and 6 p.m. In addition, the rate of transportation usage peaked at 8 a.m. and 7 p.m. during weekdays. The participants' time spent at home increased, and working hours significantly decreased on weekends. The time spent on leisure, other indoor events, outdoor occasions, and transportation was centralized at around 3 p.m.

**Table 2.** Time spent (h) indoors, outdoors, and in transportation.

| Microenvironment | Weekday | | | Weekend | | | Total | | |
|---|---|---|---|---|---|---|---|---|---|
| | Mean ± SD | Rate (%) | Sum (%) | Mean ± SD | Rate (%) | Sum (%) | Mean ± SD | Rate (%) | Sum (%) |
| Indoor | | | 84.08 | | | 84.72 | | | 84.26 |
| Home | 13.18 ± 3.20 | 54.92 | | 16.71 ± 4.27 | 69.61 | | 14.19 ± 2.87 | 59.12 | |
| Workplace | 5.04 ± 3.37 | 20.98 | | 1.11 ± 2.51 | 4.60 | | 3.91 ± 2.60 | 16.30 | |
| Leisure | 0.33 ± 0.68 | 1.38 | | 0.28 ± 0.89 | 1.15 | | 0.31 ± 0.56 | 1.31 | |
| Restaurant/bar | 0.61 ± 0.56 | 2.54 | | 0.59 ± 0.74 | 2.46 | | 0.60 ± 0.52 | 2.52 | |
| Others | 1.02 ± 2.11 | 4.26 | | 1.66 ± 2.74 | 6.90 | | 1.25 ± 2.15 | 5.01 | |
| Outdoor | | | 7.60 | | | 9.15 | | | 8.04 |
| Home | 0.01 ± 0.06 | 0.05 | | 0.09 ± 0.44 | 0.38 | | 0.03 ± 0.14 | 0.14 | |
| Workplace | 0.26 ± 0.83 | 1.09 | | 0.05 ± 0.27 | 0.22 | | 0.20 ± 0.63 | 0.84 | |
| Leisure | 0.26 ± 0.62 | 1.09 | | 0.47 ± 0.84 | 1.95 | | 0.32 ± 0.58 | 1.34 | |
| Restaurant/bar | 0.15 ± 0.30 | 0.60 | | 0.19 ± 0.65 | 0.79 | | 0.15 ± 0.36 | 0.66 | |
| Others | 0.15 ± 1.00 | 4.77 | | 1.40 ± 1.37 | 5.81 | | 1.22 ± 0.94 | 5.07 | |
| Transportation | | | 8.32 | | | 6.14 | | | 7.7 |
| Bus | 0.51 ± 5.89 | 2.24 | | 0.28 ± 0.72 | 1.15 | | 0.46 ± 0.51 | 1.93 | |
| Subway | 0.75 ± 0.80 | 3.14 | | 0.31 ± 0.59 | 1.28 | | 0.63 ± 0.60 | 2.61 | |
| Taxi | 0.03 ± 0.09 | 0.13 | | 0.02 ± 0.09 | 0.09 | | 0.03 ± 0.07 | 0.12 | |
| Car | 0.59 ± 0.99 | 2.45 | | 0.82 ± 1.04 | 3.40 | | 0.65 ± 0.89 | 2.72 | |
| Others | 0.09 ± 0.25 | 0.36 | | 0.05 ± 0.25 | 0.22 | | 0.08 ± 0.24 | 0.32 | |

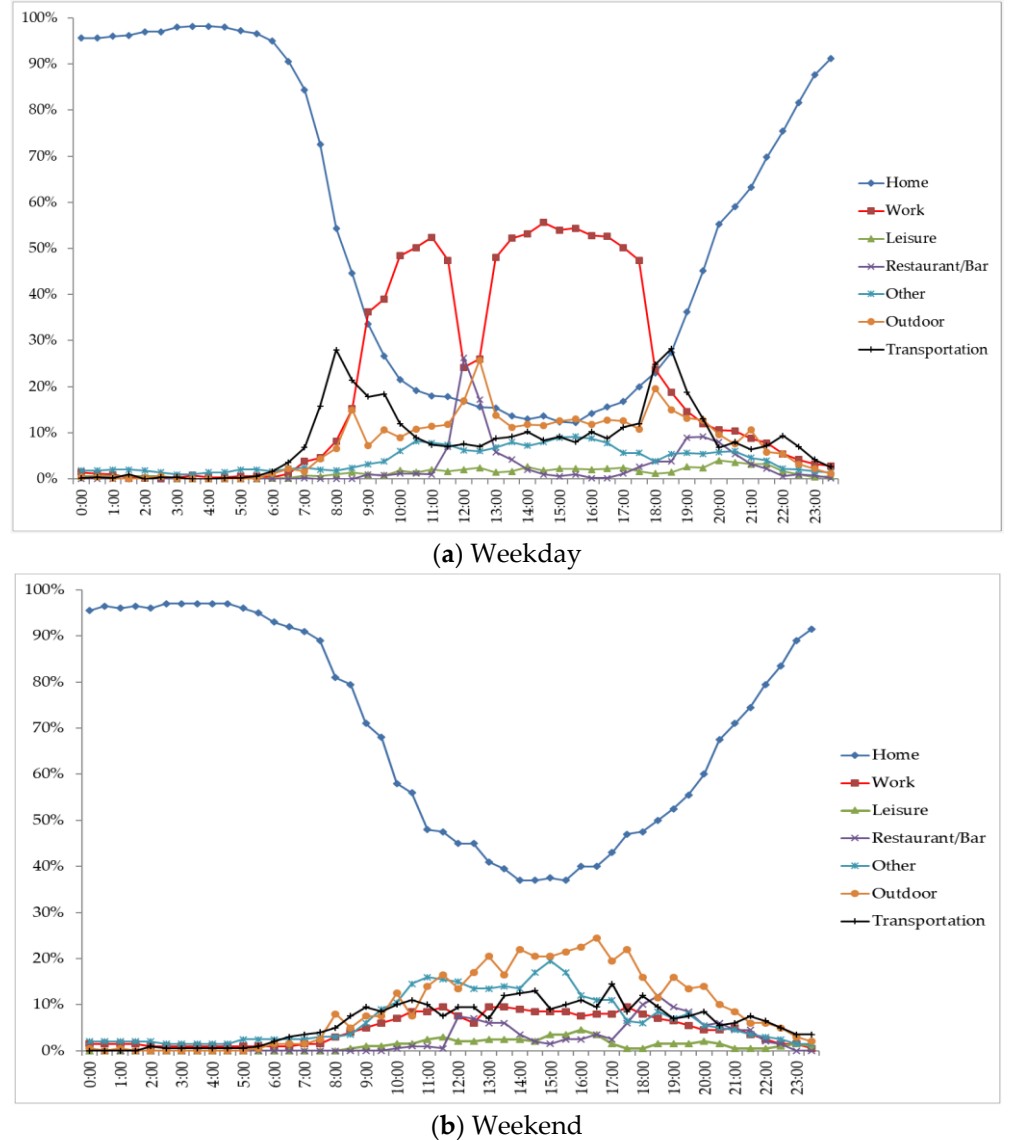

(**a**) Weekday

(**b**) Weekend

**Figure 2.** Percentage of participants at each microenvironment by the time of day for 24 h in weekdays and weekends.

### 3.4. SHS Exposure Pattern

Figure 3 shows the number of smokers observed, the number of cigarette butts found, and the number of self-reported exposures to SHS by time for 7 days. The number of cigarette butts, smokers, and exposure to SHS increased during commuting (8 a.m. and 6 p.m.) and lunch hours (12 p.m. to 1 p.m.) during weekdays. During the weekend, the number of cigarette butts found was highest between 10 a.m. and 11 a.m., and the second highest number was observed at 2 p.m. The number of exposures to SHS was consistently distributed between 8 a.m. and 11 p.m., and the number of smokers observed was divided into two periods (3 a.m.–12 p.m. and 4 p.m.–11 p.m.).

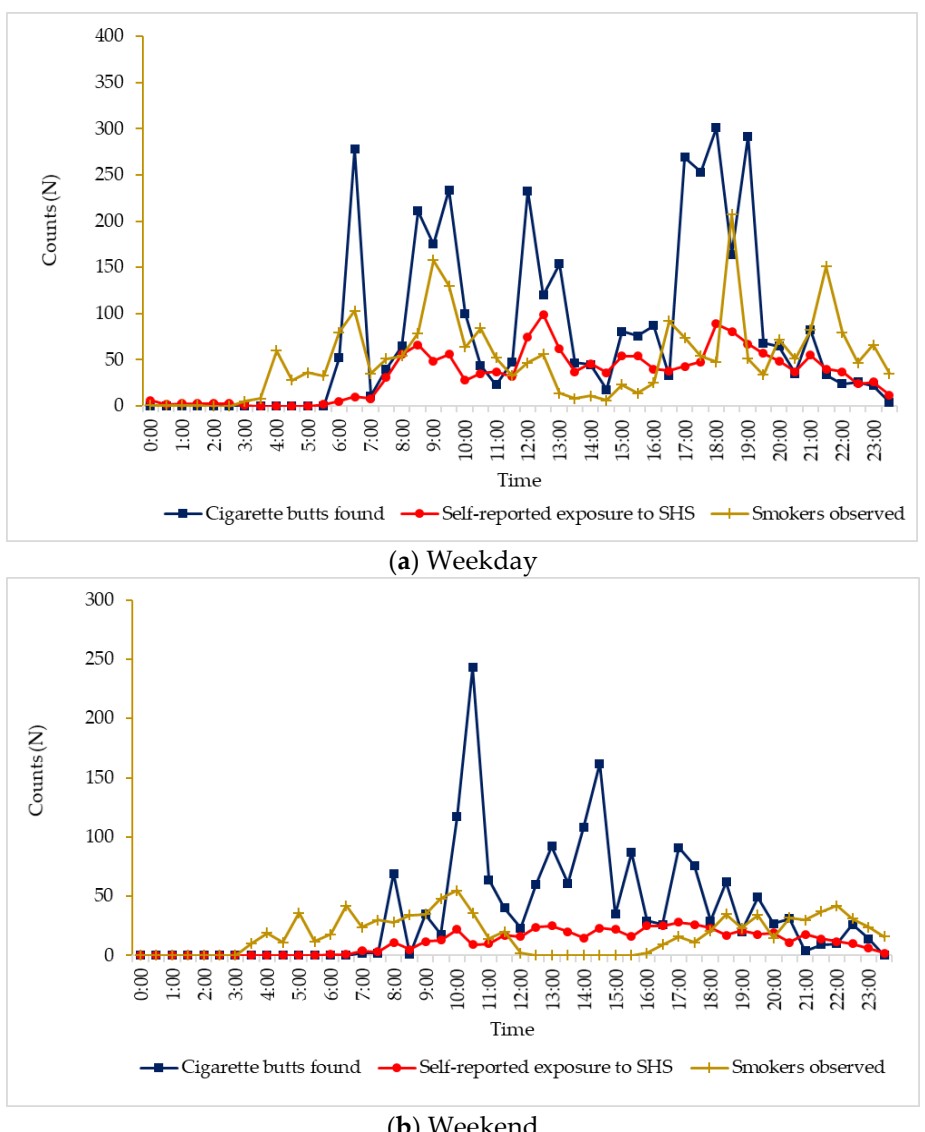

(**a**) Weekday

(**b**) Weekend

**Figure 3.** The number of cigarette butts found, smokers observed, and self-reported exposure to SHS during weekdays and weekends.

### 3.5. Time Activity Pattern and SHS Exposure Pattern

The results of the correlation analysis between time activity patterns and SHS exposure are presented in Table 3. The number of cigarette butts found and the number of self-reported SHS exposures significantly decreased as the time spent indoors increased. However, there was a statistically significant positive relationship between the time spent at work, leisure, restaurant, other indoor spaces, and the number of cigarette butts found and the number of self-reported SHS exposures. In particular,

the number of cigarette butts found and self-reported SHS exposures were significantly correlated with the time spent outdoors (Figure 4).

**Table 3.** Correlation between the time spent at each microenvironment and number of cigarette butts found, smokers observed, and self-reported SHS exposure by time.

| | Weekday | | | Weekend | | |
|---|---|---|---|---|---|---|
| **Time Spent** | **Cigarette Butts Found** | **Self-Reported Exposure to SHS** | **Smokers Observed** | **Cigarette Butts Found** | **Self-Reported Exposure to SHS** | **Smokers Observed** |
| Indoor | | | | | | |
| Home | −0.519 ** | −0.783 ** | −0.212 | −0.637 ** | −0.916 ** | 0.126 |
| Workplace | 0.321 * | 0.494 ** | 0.060 | 0.684 ** | 0.864 ** | −0.100 |
| Leisure | 0.252 | 0.638 ** | 0.210 | 0.450 ** | 0.720 ** | −0.232 |
| Restaurant/bar | 0.371 ** | 0.590 ** | 0.047 | 0.212 | 0.656 ** | −0.002 |
| Others | 0.248 | 0.572 ** | 0.040 | 0.678 ** | 0.719 ** | −0.217 |
| Outdoor | 0.605 ** | 0.935 ** | 0.261 | 0.561 ** | 0.930 ** | −0.191 |
| Transportation | 0.621 ** | 0.766 ** | 0.527 ** | 0.672 ** | 0.867 ** | 0.108 |

* $p < 0.05$; ** $p < 0.01$.

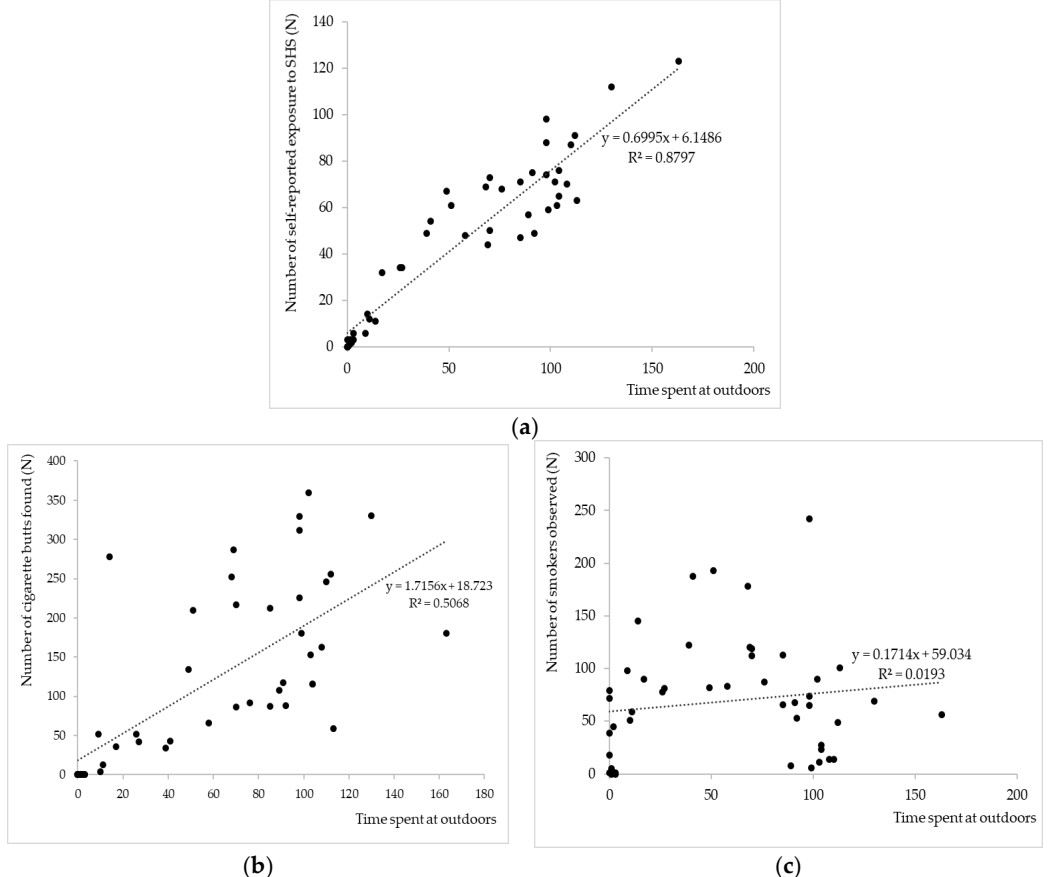

**Figure 4.** Correlation between time spent outdoors and (**a**) self-reported SHS exposure; (**b**) cigarette butts found; and (**c**) smokers observed for 7 days.

### 3.6. Biological Levels

The concentrations of cotinine, NNAL, and nicotine are presented in Table 4. Urine samples for cotinine and NNAL were corrected by measuring concentrations of creatinine in urine to minimize physiological reactions caused by kidney diseases. Urine samples for cotinine and NNAL were collected twice with weekly intervals to determine the half-life. However, there was no significant difference between the pre- and post-sampling results. Additionally, 64–69% of cotinine, 75% of NNAL, and 78% of hair nicotine samples were higher than the LOQ and were considered to be exposed to SHS.

However, the samples that indicated the participants' exposure to SHS were less than 10% when using cutoffs from published literature [21,37,38]. The results showed a statistically significant relationship between the analytes (Table 5).

**Table 4.** Concentrations of cotinine and 4-(methylnitrosamino)-1-3-(pyridyl)-1-butanol (NNAL) in urine and nicotine in hair.

| N = 100 | GM [1] (GSD) [2] | 95% CI | *p*-Value [3] | SHS Exposure Criteria | |
|---|---|---|---|---|---|
| | | | | >LOQ [4] (%) | >Cut-Off [5] (%) |
| Cotinine1 (ng/mL) | 0.58 (4.72) | 0.43–0.79 | 0.508 | 64 | 6 |
| Cotinine2 (ng/mL) | 0.53 (3.86) | 0.40–0.69 | | 69 | 7 |
| NNAL1 (pg/mL) | 1.12 (2.93) | 0.91–1.39 | 0.397 | 75 | 9 |
| NNAL 2(pg/mL) | 1.04 (2.62) | 0.86–1.26 | | 75 | 8 |
| Nicotine, hair (ng/mg) | 0.09 (2.63) | 0.07–0.10 | | 79 | 0 |

[1] Geometric mean, [2] Geometric standard deviation, [3] Paired t-test, [4] Limit of quantification, [5] cut-off for cotinine, 5 ng/mL; NNAL, 3.77 pg/mL; and hair nicotine, 2 ng/mg.

**Table 5.** Correlations between cotinine, NNAL and nicotine levels.

| | Cotinine 1 | NNAL 1 | Cotinine 2 | NNAL 2 | Nicotine in Hair |
|---|---|---|---|---|---|
| Cotinine 1 | 1 | | | | |
| NNAL 1 | 0.412 ** | 1 | | | |
| Cotinine 2 | 0.537 ** | 0.443 ** | 1 | | |
| NNAL 2 | 0.381 ** | 0.646 ** | 0.519 ** | 1 | |
| Nicotine in hair | 0.335 ** | 0.339 ** | 0.341 ** | 0.455 ** | 1 |

** $p < 0.01$.

### 3.7. Biological Levels and Time Activity Pattern

Correlation analysis between the participants' time spent in each microenvironment for 7 days and their biological sampling results are presented in Table 6. NNAL concentrations showed a marginal negative correlation with the time spent indoors at home, whereas the NNAL2 concentration had a positive relationship with the time spent at other transportations. The nicotine concentration in hair showed a statistically significant negative correlation with the time spent at indoor leisure facilities. However, a statistically significant positive correlation was observed between the hair nicotine concentration and the time spent at outdoor leisure facilities. Cotinine concentrations did not show any significant correlation with the time spent in each microenvironment.

**Table 6.** Correlations between biological sampling results and time spent in each microenvironment.

| Variables | Cotinine 1 | NNAL 1 | Cotinine 2 | NNAL 2 | Nicotine in Hair |
|---|---|---|---|---|---|
| Indoor | | | | | |
| Home | −0.070 | −0.281 ** | −0.025 | −0.228 * | −0.073 |
| Work | 0.129 | 0.142 | 0.117 | 0.070 | −0.050 |
| Leisure | 0.047 | −0.063 | −0.117 | 0.015 | −0.199 * |
| Restaurant/bar | 0.123 | 0.022 | 0.109 | 0.072 | 0.005 |
| Others | −0.094 | 0.094 | −0.185 | 0.109 | 0.121 |
| Outdoor | | | | | |
| Home | −0.054 | −0.124 | −0.051 | −0.021 | 0.021 |
| Work | −0.039 | −0.013 | 0.087 | 0.001 | −0.037 |
| Leisure | 0.152 | 0.094 | 0.075 | 0.162 | 0.204 * |
| Restaurant/bar | −0.033 | 0.013 | 0.043 | 0.117 | 0.079 |
| Others | 0.021 | 0.022 | −0.005 | −0.117 | −0.003 |
| Transportation | | | | | |
| Bus | 0.086 | 0.089 | 0.013 | 0.152 | 0.019 |
| Subway | −0.072 | 0.099 | −0.055 | 0.122 | 0.107 |
| Taxi | −0.027 | 0.001 | 0.022 | 0.135 | −0.080 |
| Car | −0.104 | 0.087 | 0.075 | −0.066 | −0.036 |
| Others | 0.036 | 0.078 | 0.136 | 0.264 ** | 0.100 |

* $p < 0.05$; ** $p < 0.01$.

*3.8. Biological Levels and SHS Exposure Indices*

The results of the correlation analysis between biological levels and SHS exposure indices are presented in Table 7. There was a marginal positive relationship between the concentration of cotinine and the number of smokers observed and self-reported SHS exposure. In addition, NNAL had a marginal positive relationship with the number of smokers observed. There was no significant relationship between the number of cigarette butts found with any biological sample and the concentration of hair nicotine and any of the SHS exposure indices.

**Table 7.** Correlations between self-reported SHS exposure indices and biological exposure indices.

| Variables | Cotinine 1 | NNAL 1 | Cotinine 2 | NNAL 2 | Nicotine in Hair |
|---|---|---|---|---|---|
| Number of cigarette butts found | 0.049 | 0.042 | 0.103 | 0.081 | 0.065 |
| Number of smokers observed | 0.207 * | 0.255 * | 0.233 * | 0.331 ** | 0.157 |
| Number of self-reported SHS exposure | 0.220 * | 0.147 | 0.306 ** | 0.19 | 0.091 |

* $p < 0.05$; ** $p < 0.01$.

## 4. Discussion

Considering the health effects of SHS, various policies, including the prohibition of indoor smoking, are being implemented worldwide. The pattern of SHS exposure is changing from indoor to outdoor environments. Therefore, this study assessed the degree of SHS exposure in outdoor environments, taking into account the pattern of people's time activities. The methodology used in this study and the results derived might be used in non-smoking policy.

The results of SHS exposures were compared with previous study conducted several countries. A total of 97% of the participants responded that they had been exposed to SHS at least once during the last week. The SHS exposure rates were 92% for outdoors, 62% at public places, 39% at workplaces, and 36% at home. In comparison with a study by Eriksen et al., SHS exposure rates were the highest after China, Bangladesh, Egypt, Vietnam, Greece, and Indonesia [9]. These SHS exposure rates were higher than the Korea National Health and Nutrition Examination Survey (home: 4.7%, workplaces: 12.7%, and public places: 21.1%) [22]. The results indicate that SHS exposure rates obtained by a simple questionnaire may be subjective and overestimated depending on the assessment method. Non-smokers can perceive not only the smell of cigarettes directly, but also smokers' breathing and body odor as SHS exposure. Therefore, it is necessary to clarify the route of SHS exposure.

Based on the questionnaire, the microenvironment where the participants were most exposed to SHS was outdoors. Kaufman et al. and Sureda et al. have reported that smoking areas are shifting from indoors to outdoors [16,20]. In the results of the time activity survey conducted in this study, participants spent approximately 8% of their daily time outdoors. Time activity studies previously conducted by Klepis et al. and Yang et al. indicated that people spent 5% of their time outdoors, which is approximately 1 to 2 h per day [39–41]. As the time spent outdoors was relatively short, the effectiveness of the outdoor smoking ban has been widely debated [42,43]. However, Lopez et al. reported that the concentration of nicotine and other substances from SHS in the air can be higher in terraces or corridors of buildings with a smoking ban, as compared to their concentrations outdoors. In addition, building types and ventilation conditions could also affect the concentration of airborne SHS indicators [20,44]. Therefore, building users may be intermittently exposed to high concentrations of SHS when they enter or exit buildings, which may have a negative psychological effect.

The number of cigarette butts found, smokers observed, and self-reported SHS exposure showed similar patterns over time, which were also similar to the participants' time activity patterns for outdoors. This indicates that the participants were frequently exposed to SHS outdoors while moving

from one space to another. Several studies have reported that exposure at entrances and exits of buildings has been an issue, and some countries are expanding non-smoking areas to these outdoor locations [16,18,20,44].

The classification of non-smokers exposed to SHS, based on existing literature, showed that most of their exposures to SHS were low. However, according to the classification by CLSI, 64–78% of non-smokers were exposed to SHS. Therefore, further studies are required to set the criteria for biological samples for SHS exposure. In addition, Hecht et al. reported that concentrations of NNAL and NNAL-Gluc in some human subjects could be detected even after 281 days since smoking cessation [45]. Therefore, these factors should be considered when selecting non-smokers in further SHS exposure assessment studies.

The number of smokers observed during a week showed a marginal correlation with the concentrations of cotinine and NNAL, and the number of self-reported exposures to SHS had a significant positive correlation with the concentrations of cotinine. Although the correlations between time spent at each location and concentrations of biomarkers were marginal, the results showed that the time activity and SHS exposure patterns could be individually used to assess non-smokers' exposure to SHS. As most residential areas prohibit smoking, non-smokers would have a lower chance of SHS exposure as they stay at home for longer durations. Likewise, time spent at other indoor areas would also decrease SHS exposure and biomarker concentrations. However, non-smokers' SHS exposure and biomarker concentrations would be higher if they frequently move within different microenvironments, which would increase the time spent outdoors. In addition, their exposure to SHS would increase as smokers would smoke at entranceways, exits, or terraces of buildings with a smoking ban [46]. There was no significant correlation between the concentration of cotinine and time activities. This can be explained by the half-life of cotinine being 18–24 h, which cannot be compared with weekly activities [47].

In this study, the main microenvironment for SHS exposure was outdoors, especially entrances to restaurants, bars, clubs, and places related to transportation. In addition, it was also found that people were mainly exposed to SHS during peak movement hours, such as rush hour while commuting and lunch hours. In Korea, smoking is prohibited indoors; however, smokers can be seen in areas near entrances and exits of buildings such as restaurants, bars, malls, and clubs. Even though the duration of SHS exposure in such a microenvironment was relatively low, the facility users could be exposed to high concentrations of SHS, causing discomfort. Therefore, management and countermeasures are required to control the exposure to SHS in such microenvironments. The number of participants in this study was relatively low, and the types of public places visited by each participant were limited. In addition, their time spent in some of the microenvironments was very short. Hence, there were limitations in analyzing the relationship between the time spent in every microenvironment and the respective concentrations of biomarkers. Therefore, in future studies, a longer-term study period and larger-scale study of the population would be required to obtain a detailed exposure assessment and the relationship between the times spent in microenvironments and the concentrations of biomarkers for SHS exposure. The sample size and of 100 non-smokers may not be representative to assess the exposure to SHS. However, this study suggested the methodology by combining questionnaires and biomarkers with time activity patterns. The results of this study could be used as a basis for establishing an expansion to the non-smoking area policies, designation of outdoor smoking areas, non-smoking while walking, and outdoor smoking during rush hour, which would help reduce SHS exposure.

## 5. Conclusions

While indoor smoking is decreasing worldwide, outdoor smoking is increasing. Thus, non-smokers can still be exposed to SHS. The SHS exposure assessment using time activity patterns can be used to identify non-smokers' temporal and spatial characteristics of SHS exposure. This method can also be used to establish management priorities. In this study, the main microenvironments where non-smokers are exposed to SHS were outdoors, mainly related to public places such as restaurants,

bars, clubs, entrances, and transportation. In addition, the participants experienced exposure to SHS during commuting time and lunch hours. Non-smokers may have more chances to be exposed to SHS if they visit any of these places frequently. Therefore, time activity surveys need to be an essential factor in future SHS exposure assessment studies. The data from these studies could be used to identify detailed information on time and the microenvironment of SHS exposure to assess long-term SHS exposures. The study can also help reinforce smoking bans at entrances and surrounding spaces around buildings such as restaurants, bars, and clubs.

**Author Contributions:** Data collection and writing—original draft, B.L.W.; supervision, conceptualization, review, and editing, M.K.L. and E.Y.P.; investigation, J.P., H.R. and D.J.; review and editing, M.J.R.; supervision, conceptualization, writing, review, and editing, W.Y. All authors have read and agreed to the published version of the manuscript.

**Funding:** This research was funded by the Korea Environmental Industry and Technology Institute (KEITI) through the Environmental Health Action Program, grant number 2018001350001.

**Conflicts of Interest:** The authors declare no conflict of interest.

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
