# Peer review of "Characteristics of Non-Smokers’ Exposure Using Indirect Smoking Indicators and Time Activity Patterns"

_sustainability, doi:10.3390/su12219099_

Round 1
Reviewer 1 Report
The article is well written but I am questioning the use of Cigar butts. Do the authors actually want to describe cigarette butts? Other articles on this topic refer to cigarettes rather than cigars. More people smoke cigarettes than cigars. Possibly this is just a typo and if so I would recommend that the text and graphs be updated before publishing this journal article. Table 1 p. 4 I am also wondering if some of the data was missing. The section on household income there were only 83 people that were included in the data there. Were the unemployed not included in this section or did some people not respond to the question? I can attach the manuscript with a few minor comments. I highlighted the Cigar butts with yellow throughout most of the manuscript.

Author Response
Thank you for your thoughtful review. We greatly appreciate your very good comments.
General comments:
The article is well written but I am questioning the use of Cigar butts. Do the authors actually want to describe cigarette butts? Other articles on this topic refer to cigarettes rather than cigars. More people smoke cigarettes than cigars. Possibly this is just a typo and if so I would recommend that the text and graphs be updated before publishing this journal article. Table 1 p. 4 I am also wondering if some of the data was missing. The section on household income there were only 83 people that were included in the data there. Were the unemployed not included in this section or did some people not respond to the question? I can attach the manuscript with a few minor comments. I highlighted the Cigar butts with yellow throughout most of the manuscript
- As for your comments and suggestions, we revised the manuscript by describing the details point by point.
Specific comments
- 1. Do the authors mean cigarette butts? Or cigar buts. It seems that less people smoke cigars than cigarettes.
- Right. We agree with your opinion. In practice, cigarette butts are common in South Korea. Therefore, we changed cigar butts into cigarette butts. .
- The household income values do not add up to 100 did some people not report for this part? Are the unemployed left out of this part? Some people may be unemployed but their household may have some income and would probably fall into the <$30,000 category.
- Thirteen unemployed and a respondent did not report to this question. We made a
- footnote that explains the difference in total number.

Reviewer 2 Report
I read the manuscript of Woo and colleagues with interest. The authors investigated an interesting research topic and wrote a good manuscript, though I believe that this paper would better fit a research letter than a complete article.
Below are some suggestions to improve the paper:
Line 27. Please correct the typing mistake: “exposed to SHS was 10% less that that found in previous studies”.
Line 27: I think that the authors wanted to say: 97% of the participants …
Line 38-39: “SHS exposure…in both smokers and non-smokers.” Please write this sentence in a clearer way, or consider removing “in both smokers and non-smokers”
Line 46-48: Please specify if these countries are developed or developing countries and add some references.
Materials and methods. “2.1. Selection of study participants”. The authors mentioned that the participants were recruited between March 22 and 28, 2017. Were the questionnaires and urine samples collected in the recruitment date (which is supposed to take place March 22 and 28, 2017)? If yes, then why the study was extended until April, knowing that the second urine sample and the activity diary were collected one week after the recruitment. Please clarify
I would like to see the questionnaire used in the study. Was it validated? Was the questionnaire developed by the authors or was it adapted from other studies? Kindly clarify
How did the authors ascertain the exposure to second-hand smoking using the questionnaire? What were the asked questions? Did previous studies use the same questions?
Line 114-115. The authors provided references about the criteria for classifying non-smokers, however it would be better to briefly explain within the main text what these criteria were.
“2.7. Statistical Analysis”: Kindly provide more details about the undertaken statistical analysis? Which type of correlation was measured? How was the statistical significance assessed?
The results section is very long. Please, revise this section and eliminate the redundant information. Consider removing the figures, or moving them to appendix, if needed.
Please rewrite the first paragraph of the discussion. Avoid repeating results and frequencies in the discussion. It would be better to start with the importance and novelty of this study and with the main finding.
Line 241, consider removing the word “significantly”
Please discuss the methodological limitations of this study such as sample size, random sampling, etc.
I think that this study needs to be repeated in a larger sample size before considering it as a basis for the non-smoking area polices. Please rewrite L301-304 while taking the sample size issue into account.
Author Response
Reviewer 2
Thank you for your thoughtful review. We greatly appreciate your very good comments.
General comments:
I read the manuscript of Woo and colleagues with interest. The authors investigated an interesting research topic and wrote a good manuscript, though I believe that this paper would better fit a research letter than a complete article. Below are some suggestions to improve the paper:
- As for your comments and suggestions, we revised the manuscript by describing the details point by point.
Specific comments
- Line 27. Please correct the typing mistake: “exposed to SHS was 10% less that that found in previous studies”
- As you pointed out, we corrected this sentence.
Line |
Before revision |
After revision |
26-27 |
The analysis of biomarker samples indicated that the proportion of participants exposed to SHS was 10% less that that found in previous studies.
|
The analysis of biomarker samples indicated that about 10% of participants were exposed to SHS when compared with the criteria from previous studies. |
- Line 27: I think that the authors wanted to say: 97% of the participants
- Right. We changed this sentence..
- Line 27: 97% of the participants
- Line 38-39: “SHS exposure…in both smokers and non-smokers.” Please write this sentence in a clearer way, or consider removing “in both smokers and non-smokers”
- As you suggested, we revised the sentence.
Line |
Before revision |
After revision |
38-39 |
SHS exposure can cause adverse health effects such as respiratory disease, cardiovascular disease, and lung cancer in both smokers and non-smokers [2,3]
|
SHS exposure can cause adverse health effects such as respiratory disease, cardiovascular disease, and lung cancer [2,3]. |
- Line 46-48: Please specify if these countries are developed or developing countries and add some references.
- As your suggestion, we added references and revised this sentence. .
Line |
Before revision |
After revision |
46-48 |
In accordance with the FCTC, several countries have successfully implemented smoke-free policies in indoor environments such as homes, workplaces, and public places during the last decade.
|
According to the FCTC, fifteen developed and developing countries have successfully implemented smoke-free policies in indoor environments such as homes, workplaces, and public places during the last decade [9,10]. |
- Materials and methods. “2.1. Selection of study participants”. The authors mentioned that the participants were recruited between March 22 and 28, 2017. Were the questionnaires and urine samples collected in the recruitment date (which is supposed to take place March 22 and 28, 2017)? If yes, then why the study was extended until April, knowing that the second urine sample and the activity diary were collected one week after the recruitment. Please clarify
- We made a mistake in describing the period. The recruitment date was different from the actual survey date. There was a difference in the duration of subjects’ participation in this study. Therefore, we modified this sentence as follows.
Line |
Before revision |
After revision |
77 |
Among the participants recruited for the basic SHS exposure survey conducted by the National Cancer Center, 100 non-smokers aged 19 to 74 and residing in Seoul were randomly selected between March 22, and 28, 2017. |
Among the participants recruited for the basic SHS exposure survey conducted by the National Cancer Center, 100 non-smokers aged 19 to 74 and residing in Seoul were randomly selected. Questionnaires, time activity diaries, and biomarker samples were collected from April to May 2017. |
- I would like to see the questionnaire used in the study. Was it validated? Was the questionnaire developed by the authors or was it adapted from other studies? Kindly clarify. How did the authors ascertain the exposure to second-hand smoking using the questionnaire? What were the asked questions? Did previous studies use the same questions?
- We understood what you mean. Basically, we can make use of various methods when we study the SHS researches, but each can have advantages and disadvantages. Among them, using questionnaire can be the most subjective. However, as you know, it can be applied to many people at low costs. In conclusion, we have approached second-hand smoke in various ways in this study. The limitations of the questionnaire were described in the Discussion Section, and the questionnaire used was adapted from the previous studies.
- → As you pointed out, “The used questionnaire was adapted from the previous studies [15,23-27]” was inserted.
- Line 114-115. The authors provided references about the criteria for classifying non-smokers, however it would be better to briefly explain within the main text what these criteria were.
- The meaning of this sentence may be not clear. It means that once the results of biomarker analysis were detected above LOQ, they were classified as exposed to SHS. Therefore we revised sentence like below to clarify.
Line |
Before revision |
After revision |
114-115 |
The criteria for classifying exposure to SHS of non-smokers were established by reviewing literature and articles suggested by several organizations [19,30,31].
|
The criteria for classifying exposure to SHS of non-smokers were established by reviewing literature and articles suggested by several organizations [21,37,38]. The non-smokers were classified to be exposed SHS when the concentration above LOQ was detected as the results of biomarker analysis, or according to cut-off concentration (1 ng/mL, 3.77 pg/mL, and 2 ng/mg, cotinine in urine, NNAL in urine and nicotine in hair respectively). The limit of quantitation (LOQ) of cotinine, NNAL, and nicotine were obtained according to the Clinical & Laboratory Standards Institute (CLSI) guidelines.
|
- “2.7. Statistical Analysis”: Kindly provide more details about the undertaken statistical analysis? Which type of correlation was measured? How was the statistical significance assessed?
- As you suggested, we added a sentence.
Line |
Before revision |
After revision |
121-124 |
Correlation was used to compare the time activity patterns, concentrations of biomarkers, number of cigarette butts found, number of smokers observed, and the number of exposures to SHS. IBM SPSS (Version 19) was used for statistical analysis.
|
Correlation was used to compare the time activity patterns, concentrations of biomarkers, number of cigarette butts found, number of smokers observed, and the number of exposures to SHS. The results of correlation analysis were presented by coefficient of correlation (R2) and statistical analysis was performed using IBM SPSS (Version 19).
|
- The results section is very long. Please, revise this section and eliminate the redundant information. Consider removing the figures, or moving them to appendix, if needed.
- We agree with you opinion. However, we think that both the contents and the figures of the Results Section are necessary. Please understand that it is for the understanding of the main results of this study and other researchers.
- Please rewrite the first paragraph of the discussion. Avoid repeating results and frequencies in the discussion. It would be better to start with the importance and novelty of this study and with the main finding.
- Thanks for your suggestion. We revised the first paragraph of the Discussion Section.
Line |
Before revision |
After revision |
|
|
Considering the health effects of SHS, various policies including the prohibition of indoor smoking are being implemented worldwide. The pattern of SHS exposure is changing from indoor to outdoor environments. Therefore this study assessed the degree of SHS exposure in outdoor environments taking into account the pattern of people’s time activities. The methodology used in this study and the results derived might be used in non-smoking policy. The results of SHS exposures were compared with previous study conducted several countries. |
- Line 241, consider removing the word “significantly”
- As you suggested, we removed this word.
- Please discuss the methodological limitations of this study such as sample size, random sampling, etc. I think that this study needs to be repeated in a larger sample size before considering it as a basis for the non-smoking area polices. Please rewrite L301-304 while taking the sample size issue into account.
- As your opinion, this study has limitations to assess exposure to SHS because of the size of subjects. However, please consider that the purpose of this study did not assess the exposure to SHS in some population, but suggesting methodology to SHS exposure assessment by combining questionnaires and biomarkers with time activity patterns for detailed exposure assessment. So we emphasized these points. Therefore, we revised.
Line |
Before revision |
After revision |
301-304 |
The results of this study could be used as a basis for establishing an expansion to the non-smoking area policies, designation of outdoor smoking areas, non-smoking while walking, and outdoor smoking during rush hour, which would help reduce SHS exposure [42].
|
The sample size of 100 non-smokers may not be representative to assess the exposure to SHS. However, this study suggested the methodology by combining questionnaires and biomarkers with time activity patterns. The results of this study could be used as a basis for establishing an expansion to the non-smoking area policies, designation of outdoor smoking areas, non-smoking while walking, and outdoor smoking during rush hour, which would help reduce SHS exposure. |
